# An Overview of Multi-Criteria Decision Analysis (MCDA) Application in Managing Water-Related Disaster Events: Analyzing 20 Years of Literature for Flood and Drought Events

**Mohammad Fikry Abdullah** [1,2,*] , **Sajid Siraj** [1] **and Richard E. Hodgett** [1]

1   Leeds University Business School, University of Leeds, Leeds LS2 9JT, UK; S.Siraj@leeds.ac.uk (S.S.); R.E.Hodgett@leeds.ac.uk (R.E.H.)
2   National Water Research Institute of Malaysia (NAHRIM), Seri Kembangan Selangor 43300, Malaysia
*   Correspondence: fikry@nahrim.gov.my or fikry.abdullah@gmail.com

**Abstract:** This paper provides an overview of multi-criteria decision analysis (MCDA) applications in managing water-related disasters (WRD). Although MCDA has been widely used in managing natural disasters, it appears that no literature review has been conducted on the applications of MCDA in the disaster management phases of mitigation, preparedness, response, and recovery. Therefore, this paper fills this gap by providing a bibliometric analysis of MCDA applications in managing flood and drought events. Out of 818 articles retrieved from scientific databases, 149 articles were shortlisted and analyzed using a Preferred Reporting Items for Systematic Reviews and Meta-analyses (PRISMA) approach. The results show a significant growth in MCDA applications in the last five years, especially in managing flood events. Most articles focused on the mitigation phase of DMP, while other phases of preparedness, response, and recovery remained understudied. The analytical hierarchy process (AHP) was the most common MCDA technique used, followed by mixed-method techniques and TOPSIS. The article concludes the discussion by identifying a number of opportunities for future research in the use of MCDA for managing water-related disasters.

**Keywords:** multi-criteria decision analysis (MCDA); water-related disaster; floods; drought; disaster management plan (DMP); systematic literature review (SLR); PRISMA

## 1. Introduction

Water-related disaster (WRD) events create complex problems, and solving these problems requires knowledge and expertise from various disciplines, including the environmental, economic, and social domains. Due to its multidisciplinary nature, multi-criteria decision analysis (MCDA) is a popular way to analyze these problems. In this paper, we first introduce WRDs and MCDA, followed by an analysis of MCDA techniques in managing disasters, explicitly focusing on flood and drought events.

A recent report from the United Nations Office for Disaster Risk Reduction (UNISDR) stated about 90% of all natural disasters are WRDs. These disasters are caused by environmental control (climate variability), management control (inappropriate land use), and socio-economic pressure (development and construction in high-risk areas) [1]. The occurrences of various types of natural disasters from 1995 to 2014 are as follows: flood, 43%; storm, 28%; earthquake, 8%; extreme temperature, 6%; landslide, 5%; drought, 5%; wildfire, 4%; and volcanic activity, 2% [2]. The impact of WRDs includes economic and environmental damages, fatalities, reconstruction costs, aesthetic damage, disruption of normal activities, loss of assets, and long-term or permanent loss of species [3]. For example, social impacts include loss of life, population displacement, and other adverse effects such as those on physical, mental, and social well-being. Losses and damages to infrastructure and assets and discontinuation of services are among the impacts on the economy. During the 2000–2010 period, Gopalakrishnan reported that WRDs resulted in

87% of all total disasters, with 37% of the fatalities and 76% of the overall economic losses from all disasters combined [4]. Flood and drought events have caused damages that cost more than USD 0.4 billion, as recorded by the Centre for Research on the Epidemiology of Disasters (CRED) [5]. Based on various studies, the impact of climate change, together with improper land use planning, will amplify both events in frequency and intensity in the future, which will cause more significant impacts on the economy, environment, and human life [6,7].

From a management perspective, a disaster management plan (DMP) can be categorized into four phases: mitigation, preparedness, recovery, and response [8,9] (these phases are introduced and discussed in the next section in detail). In each DMP phase, the objective to achieve is different, thus requiring a different set of criteria for a better actionable plan and subsequent decision. However, criteria are often conflicting with each other. For example, several social and environmental criteria may require high costs and too many resources, which might not be possible due to financial and technical constraints. Therefore, it is crucial to find the best possible trade-offs or compromises between these conflicting criteria and objectives. In this context, MCDA offers techniques that support decision-making when there are multiple conflicting criteria involved. They are practical and powerful tools that facilitate the scientific integration of quantitative and qualitative analyses [9]. MCDA has been used in various applications to manage natural disasters, such as in resilience index estimation [10], flood hazard assessment [11], risk index estimation [12], and policy development [13].

This study aims to review the literature and perform a critical assessment of MCDA techniques used in flood and drought management. It provides in-depth information on MCDA techniques for both events and identifies new potential research opportunities according to the DMP phases for disaster management.

This paper is structured as follows: Section 2 describes this study's background; Section 3 explains the methodology; Section 4 reviews MCDA techniques for the management of flood and drought events. Criticism of existing literature and future potential research is provided in Section 5, followed by an overview of the conclusions in Section 6.

## 2. Study Background

The term "disaster" refers to an event causing destruction and suffering. More formal definitions are provided by the Centre for Disaster Epidemiology Research (CRED) [14] and the United Nations Office for Disaster Risk Reduction (UNDRR) [9]. These are provided below for reference:

Definition by CRED: "*a situation or event that overwhelms local capacity, requiring a request for external assistance at the national or international level where an unforeseen and often sudden event causes significant damage, destruction, and human suffering caused by nature or human.*"

Definition by UNDRR: "*a severe disruption to the functioning of a community or society at any scale due to hazardous events interacting with conditions of exposure, vulnerability, and capacity leading to one or more of the following: human, material, economic and environmental losses, and impacts.*"

Although the first definition focuses more on destruction and suffering, the second definition has more emphasis on disruptions to communities. Nevertheless, both definitions highlight the negative impact of these disasters.

The Asian Development Bank (ADB) reported that floods and droughts are considered WRDs caused by climate change, temperature, and extreme precipitation [15].

### 2.1. Water-Related Disaster Events

Flood and drought events have been classified as the most frequent WRDs resulting in damage to the economy and properties and loss of life [1–3]. Based on the WRD events recorded by the Emergency Events Database (EM-DAT) of the Centre for Disaster Epidemiology Research (CRED) [4], there were 1906 occurrences of floods and droughts recorded worldwide between 2010 and 2020 (Table 1 shows the number of events yearly).

In this period, 1732 of these occurrences were flood events. Approximately 30% of the world's population is estimated to reside in areas routinely impacted by floods or droughts. The number of people at risk from floods alone is projected to rise to 1.6 billion in 2050 [2]. Therefore, efforts to minimize the impact and risk of floods and droughts should proactively be taken by decision-makers.

**Table 1.** Number of flood and drought events (2000–2020).

|  | 2010 | 2011 | 2012 | 2013 | 2014 | 2015 | 2016 | 2017 | 2018 | 2019 | 2020 |
|---|---|---|---|---|---|---|---|---|---|---|---|
| Flood | 184 | 156 | 136 | 148 | 137 | 161 | 159 | 126 | 128 | 196 | 201 |
| Drought | 21 | 16 | 18 | 9 | 20 | 26 | 14 | 9 | 16 | 16 | 9 |

Based on studies and reports, disaster management control measures such as the development of the risk assessment framework [5], policy-making [6], vulnerability assessment [7], and early warning systems [8] indicate the urgency of managing these impacts. A disaster management plan (DMP) offers various control measures such as decision-making processes, assessment, evaluation, policy-making, data management, and technology emergence to reduce and minimize flood and drought impacts.

### 2.2. Disaster Management Plan (DMP) Phases

There are two types of measures to minimize or prevent disaster hazards: structural and nonstructural. The United Nations Office for Disaster Risk Reduction (UN-DRR) defined structural measures as the physical construction or application of engineering/technology to achieve disaster resistance and resilience. Dams, levees, flood control reservoirs, floodwalls, rainwater harvesting systems, and runoff collection (surface and underground) are examples of structural measures. Nonstructural measures focus more on using expert knowledge, practices, or agreements to reduce disaster risks and impacts [9]. Developing flood risk maps and zoning maps, conducting assessment and evaluation, preparing projection data, and developing early warning systems are examples of nonstructural measures that decision-makers can take to minimize and prevent the impact of floods and droughts.

As a reactive measure, the four phases in DMP (mitigation, preparedness, response, and recovery) have been referred to as a process to deal with the negative effects and prevent disasters [10]. The mitigation activities focus on eliminating or minimizing disasters' impact and risk through proactive measures before a disaster happens, such as policy development, building codes, disaster-prone area identification, and vulnerability/risk assessment. The objective of the mitigation phase is long-term planning [12,13]. Preparedness activities reduce disasters' impact in advanced planning with actionable measures such as preparedness plans, emergency exercises/training, and implementation of early warning systems. The aim is short-term planning and preparation to deliver an effective response [12,13]. In the event of a disaster, immediate action is what is required in the response phase. Recovery is the final phase in which the actions and measures are taken to emphasize reconstruction and preserve life to support post-disaster continuity.

Through DMP activities, decision-makers are faced with a problem in identifying and selecting the best activities and measures to be implemented in the management of floods and droughts. Table 2 shows common problems and challenges faced by decision-makers.

Therefore, setting up objectives and identifying expected outputs or outcomes in managing drought and flood events could be supported by identifying relevant criteria using MCDA. Criteria obtained from the combination of experts' knowledge and dataset could improve flood and drought management by applying MCDA techniques.

**Table 2.** Problems and challenges in decision-making for managing disasters.

| No. | Problem and Challenges | DMP Phase | Measure Type | Example of Activities |
|---|---|---|---|---|
| 1 | Understanding disaster trends and patterns (past and future disasters) | Mitigation and Preparedness | Nonstructural | Research and assessment (development of inundation map and projection events) |
| 2 | Understanding and choosing base criteria, factors, and attributes for the DMP | Mitigation and Preparedness | Structural and Nonstructural | 1. Research and assessment (development of hazard and zoning maps); 2. Development of floodwall, dams, wetlands, drainage infrastructures, and others. |
| 3 | Development of indicators or index for disaster risk reduction | Mitigation | Structural and Nonstructural | Research and assessment (vulnerability, readiness, and adaptation index) |
| 4 | Factoring disaster risk management and setting priorities into policy development | Mitigation | Nonstructural | Research and assessment (identify, select, and rank the criteria) |
| 5 | Data management and integration (data collection, data accessibility, and data availability) limiting the capability and usability in supporting decision-making | Preparedness | Structural and Nonstructural | 1. Research and assessment (feasibility study); 2. ICT infrastructure preparedness. |

*2.3. Multi-Criteria Decision Analysis (MCDA)*

MCDA can be described as a collection of techniques for comparing, ranking, and selecting alternatives using quantifiable or nonquantifiable criteria [16,17]. MCDA has been designed to address four types of problems [18,19]:

1. The choice problem, in which MCDA is used to select the best option from a set of alternatives.
2. The sorting problem, in which MCDA is used to assign a set of alternatives to predetermined categories.
3. The ranking problem, in which MCDA is used to order the alternatives partially or completely.
4. The description problem, in which MCDA is used to define alternatives, construct a set of criteria, and determine all or some alternatives' performance for the criteria, considering additional information.

There are many academic discussions on the application of MCDA techniques in different fields and domains. Kumar [13] explained MCDA as a process for assessing real-world situations based on different qualitative/quantitative criteria in certain/uncertain/risky environments to find a suitable course of action, choice, strategy, or policy among several viable options. Zavadskas et al. [14] concluded that the implementation of MCDA is useful in helping researchers and practitioners to solve real-life problems. Velasquez and Hester [15] provide detailed guidance about implementing MCDA techniques in various social, economic, and environmental domains such as energy, water management, transportation, medical, and public policy.

The most widely used MCDA methods are described as follows:

- Analytical hierarchy process (AHP): Formulates the decision into a hierarchy of criteria and uniquely uses pairwise comparisons provided by experts' judgments to elicit preferences [20]. These preferences are then aggregated to provide recommendations.
- Analytic network process (ANP): Technique used to model a problem (hierarchic or a network structure) to represent the problem, as well as pairwise comparisons to establish relations within the structure [21].
- Data envelopment analysis (DEA): Linear programming technique used to measure the relative efficiencies of alternatives [22]. The method can be used to find the efficiency of the combination of multiple inputs and multiple outputs of a problem [20].
- Weighted sum model (WSM): A simple method for evaluating alternatives with different criteria that are expressed in the same units [20]. The value function is established based on a simple addition of scores for each alternative with respect

to each criterion, multiplied by criteria weights [23]. This method is also known as simple additive weighting (SAW).

- Weighted product model (WPM): Similar to WSM/SAW, except that multiplication is used for aggregation instead of addition [24].
- Goal programming (GP): An analytical approach devised to address decision-making problems where targets have been assigned to all the attributes and where the decision-maker (DM) is interested in minimizing the nonachievement of the corresponding goals [25].
- Elimination and choice translating reality (ELECTRE): A family of methods that utilize outranking relations to select, sort, or rank alternatives [23].
- Multi-attribute utility theory (MAUT): A methodology to incorporate risk preferences and uncertainty into multi-criteria decision support methods [26].
- Simple multi-attribute rating technique (SMART): A method very similar to WSM where all performance scores are measured/rated on a scale of 0 to 100 and then aggregated using the weighted sum approach [15].
- Preference ranking organization method for enrichment of evaluations (PROMETHEE): A family of methods that utilize outranking relations for identifying: partial ranking (I), complete ranking (II), interval ranking (III), complete or partial ranking (IV) for a continuous solution, segmentation constraints problems (V) and human brain representation (VI) [27].
- Technique for order preferences by similarity to ideal solutions (TOPSIS): Used to identify an alternative that is closest to an ideal solution and farthest from a negative ideal solution. The distances are usually measured in terms of Euclidean distance [23], although other distances are also possible.
- Simulated uncertainty range evaluations (SURE): Allows the decision-maker to provide minimum, maximum, and most likely values for each alternative with respect to each criterion. The WSM method and simulations are used to calculate distributions that represent the strength and uncertainty of each alternative [28].

Based on the strengths and weaknesses of MCDA techniques summarized in Table 3, the choice for MCDA techniques may depend on the objective and complexity of the problem. Decision-makers should consider factors such as the type of problem, decision goal, data volume, number of criteria, ease of use, consistency, and type of analysis during MCDA technique selection. This study investigates which of these techniques will work better in managing disaster events according to four phases of the disaster management plan (mitigation, preparedness, response, and recovery). This is an important research area that requires investigation to improve questions in (a) criteria identification and selection, (b) MCDA technique selection, and (c) MCDA technique application in the DMP phases. In total, 149 articles were reviewed to investigate this question. We discuss our methodology used to conduct this study in the next section.

**Table 3.** Strengths and weaknesses of MCDA techniques.

| MCDA Techniques | Strengths | Weaknesses |
|---|---|---|
| Analytic hierarchy process (AHP) [15,20,23,29] | 1. Formulates the problem into a hierarchical structure that is easy to understand and communicate;<br>2. Allows inconsistencies in judgments and comparisons. | 1. The threshold of 0.1 for rejecting inconsistent judgments remains questionable;<br>2. Prone to rank reversal;<br>3. The number of pairwise comparisons required can be high for medium to large decision problems. |
| Analytic network process (ANP) [20] | 1. Independence among elements is not required (unlike AHP);<br>2. Is considered more accurate than AHP for eliciting preferences. | 1. Even more pairwise comparisons are required than AHP from the decision-maker due to the network structure;<br>2. Uncertainty is not supported;<br>3. Difficult to understand and communicate due to complex interactions among criteria and alternatives. |
| Data envelopment analysis (DEA) [15,20,22] | 1. The relation between inputs and outputs is not necessary;<br>2. Inputs and outputs can have very different units. | 1. Measurement errors can drastically affect the model and the results;<br>2. Not suitable for large problems due to increased complexity;<br>3. Does not deal with imprecise data. |
| Weighted sum model (WSM), or simple additive weighting (SAW) [15,20] | 1. Ability to compensate among criteria (as with AHP);<br>2. Easier to comprehend due to simple arithmetic operations. | 1. Susceptible to the trap of averages [30];<br>2. All criteria must have the same units (or must be translated into the same units). |
| Weighted product model (WPM) [20] | 1. Ratios are used so there is no dependence on the unit of measurement. | 1. Does not support 0 in the weights; all criteria must have nonzero weights. |
| Goal programing (GP) [15,20,26] | 1. Handles large numbers of variables, constraints, and objectives. | 1. Setting of appropriate weights;<br>2. Needs to be combined with other MCDA methods to weight coefficients. |
| Elimination and choice translating reality (ELECTRE) [15,20,23,26] | 1. Takes uncertainty and vagueness into account;<br>2. Supports the idea of veto, which is not possible in other methods. | 1. Its process and outcomes are usually difficult to explain to nontechnical people; |
| Multi-attribute utility theory (MAUT) [15,26,29] | 1. Takes uncertainty into account. | 1. Preferences need to be precise. |
| Simple multi-attribute rating technique (SMART) [15,29] | 1. Considered simpler than other methods;<br>2. Requires less effort by decision-makers. | 1. The procedure may not be convenient considering the framework. |
| PROMETHEE [15,26,27] | 1. Supports indifference;<br>2. Supports visual aid (called GAIA). | 1. Does not provide a clear method by which to assign weights. |
| Technique for order preferences by similarity to ideal solutions (TOPSIS) [15,23] | 1. Takes into account both the best possible and worst possible options/scenarios;<br>2. Supports any form of distance measures (Euclidean, Manhattan, Chessboard, etc.). | 1. Does not consider the attributes correlation;<br>2. Each dimension has different units, so combining different dimensions needs justification. |
| Simulated uncertainty range evaluations (SURE) [28] | 1. Able to visualize the strength and uncertainty of each alternative;<br>2. Simple method to understand. | 1. The decision-maker may still need to make a choice of which alternative to select if there are many overlapping uncertainties. |

## 3. Methodology

There is a vast amount of literature on MCDA techniques and applications for natural disasters in general, and WRD has been discussed in many academic papers and government reports. Based on a broad and extensive search, this paper discusses the current development studies published between the years 2000 and 2020. The publication article selection aligns with the issues related to managing water-related disasters, specifically flood and drought events. These issues have been discussed, debated, and reported at international and national levels. The articles discussed in this paper are selected from published journals, articles presented at conferences, and articles published in proceedings.

Preferred Reporting Items for Systematic Reviews and Meta-Analyses (PRISMA) was used in this study based on its structured and well-organized process flow, which is evidence-based with minimum criteria set [16–19,31]. A comprehensive set of articles was collected based on the study setting, but some papers were likely unintentionally missed

during the search process. A thorough article search was conducted based on four phases: identification, screening, eligibility, and inclusion, as illustrated in Figure 1.

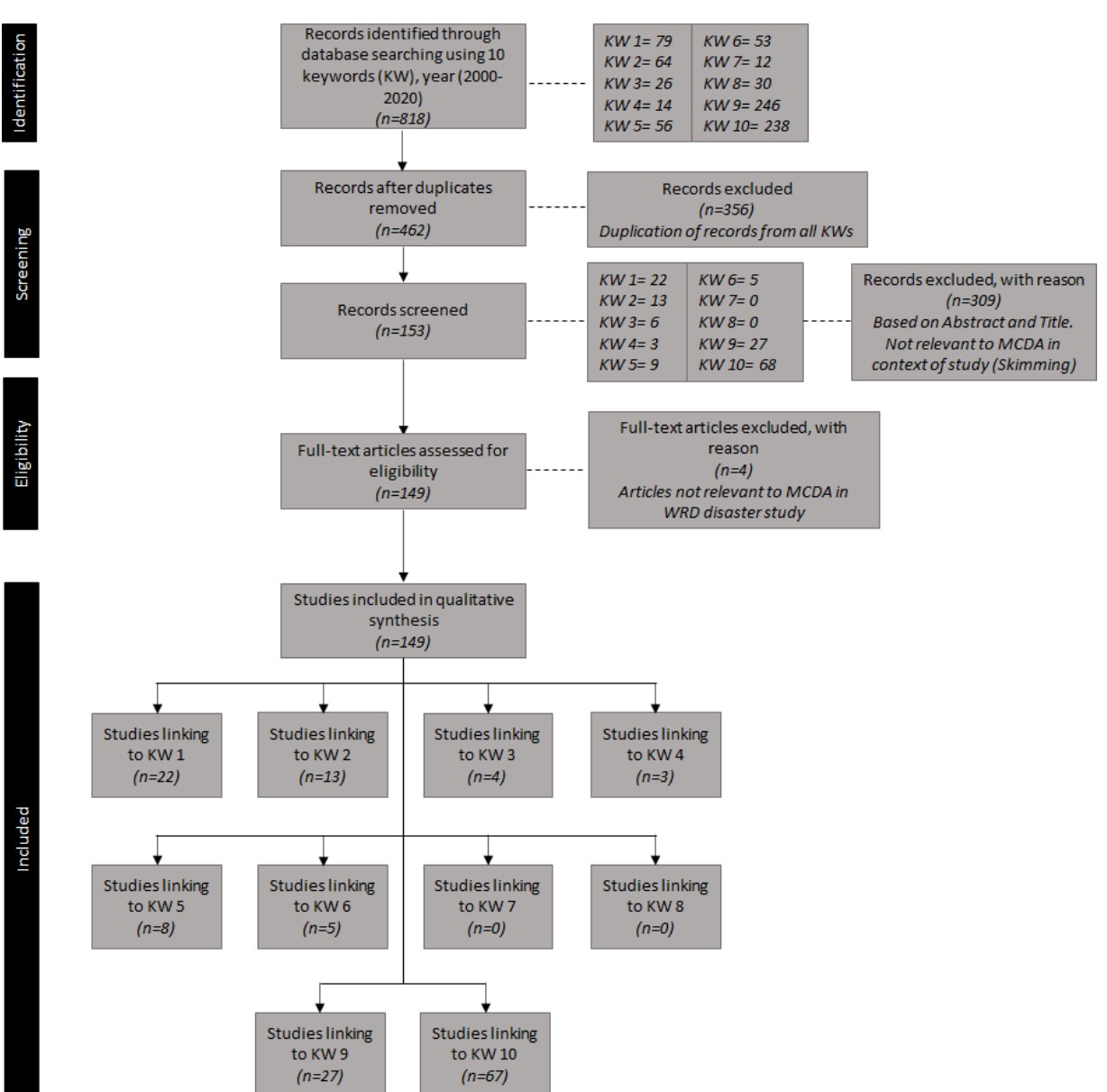

**Figure 1.** PRISMA flow diagram for the selection of eligible studies.

### 3.1. Identification of the Key Research Question

This paper aims to ascertain the current development and application of MCDA techniques in managing flood and drought events over 20 years, providing a comprehensive study to explore new research opportunities in the future. MCDA trends in managing floods and droughts can be identified by analyzing the findings based on the MCDA techniques in the DMP phases. This study explores new research opportunities in applying the new MCDA techniques, MCDA application in the DMP phases, and criteria identification and selection, which will be of benefit to the MCDA research and disaster management domains.

### 3.2. Identification of Relevant Articles

The Web of Science (WoS) database was queried using 10 combinations of keywords related to the topic. Table 4 shows the list of keywords used for the online search queries.

The keywords were selected to cover a broad investigation related to MCDM, MCDA, natural disasters, floods, and droughts.

**Table 4.** Keywords and syntax used for online article query.

| No. | Keyword | Keyword Code |
|:---:|:---:|:---:|
| 1 | "MCDM" AND "flood" | KW1 |
| 2 | "MCDA" AND "flood" | KW2 |
| 3 | "MCDM" AND "drought" | KW3 |
| 4 | "MCDA" AND "drought" | KW4 |
| 5 | "Multi-criteria decision making" AND "drought" | KW5 |
| 6 | "Multi-criteria decision analysis" AND "drought" | KW6 |
| 7 | "MCDA" AND "natural disaster" | KW7 |
| 8 | "MCDM" AND "natural disaster" | KW8 |
| 9 | "Multi-criteria decision making" AND "flood" | KW9 |
| 10 | "Multi-criteria decision analysis" AND "flood" | KW10 |

The initial online search resulted in identifying 818 articles based on the keywords in Table 4. The identification phase is shown in Figure 1 at the first phase on the top of the flow diagram. Citations of the identified articles were automatically extracted and exported into EndNote for further analysis. EndNote identified that 356 articles were duplicates; these articles were excluded, leaving 462 articles remaining.

*3.3. Selection of the Relevant Articles: Inclusion and Exclusion Criteria*

As mentioned earlier, PRISMA was used to carefully select the relevant articles for this study. Figure 1 illustrates the flow process, which summarizes the phases adopted to select the research articles eligible for detailed analysis.

In the screening phase, the titles and abstracts of the shortlisted 462 articles were manually read and assessed, out of which 309 articles were found to be out of the scope of this study, and thus only 153 remaining articles were considered for further analysis. A detailed analysis was carried out for these 153 articles by carefully examining the whole text. This detailed analysis identified four more articles that were not relevant to this study as they were not related to the use of MCDA techniques for flood and drought events; thus, they were also excluded. The final number of relevant publications used for this study was 149 articles.

*3.4. Reporting and Summarizing the Results*

As a first step, the metadata on the 149 relevant articles were extracted and compiled. This metadata included the authors' names, publication title, year of publication, MCDA techniques mentioned, the DMP phases, and the criteria. A detailed analysis of these metadata was conducted using both quantitative and qualitative approaches. The descriptive statistics were gathered to identify patterns and trends, while the qualitative and narrative approaches were used to present and discuss the results. All these analyses are presented in Sections 4 and 5 below.

## 4. Findings

The literature review identified an upward trend in using MCDA techniques for managing floods and droughts, specifically in the mitigation phase, based on their usage and popularity in decision-making. Most of the MCDA techniques have been applied as a single method; however, mixed-method techniques have increased in popularity. There is a significant gap between MCDA technique applications according to DMP phases where future works should be considered. Examples include MCDA technique selection (single or mixed-method approach), MCDA criteria selection and identification (quantity versus quality), selection of disaster events for the future case study, and MCDA application in other DMP phases.

### 4.1. Trends of Articles Based on 10 Keywords from 2000 to 2020

The keyword search shown in Figure 2 shows an upward trend of publications from 2017 to 2020. The trend of eligible articles shows the same pattern; the number of articles related to MCDA techniques for flood and drought events is expected to increase, as shown in Figure 3. In 2020, fewer articles were produced, despite the number of flood and drought occurrences increasing in 2020 (Table 1). The reason fewer articles were produced in 2020 might be due to the COVID -19 pandemic and change in the research focus. There might be a number of reasons for the decreased number of publications, which we believe is an area of further investigation.

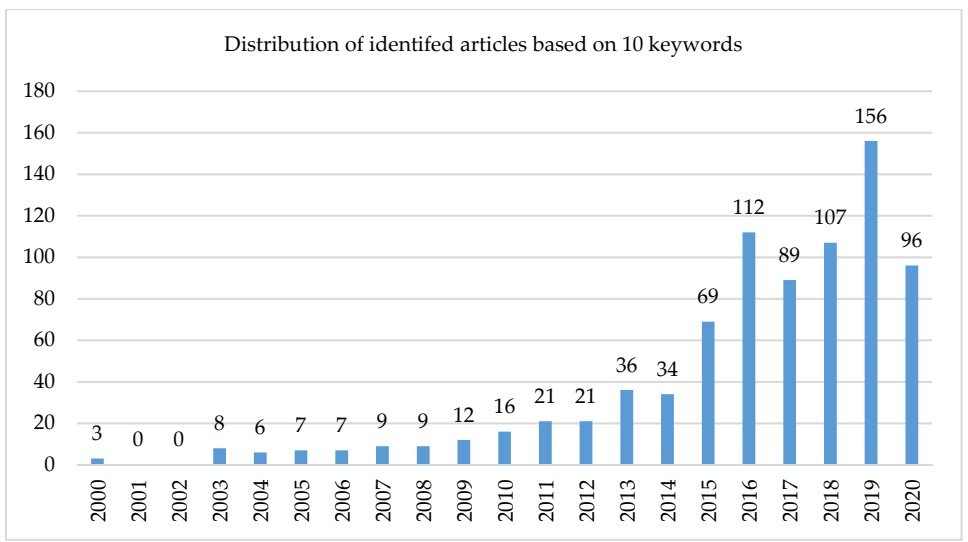

**Figure 2.** Trend of article distribution based on 818 articles (Identification Phase).

Based on the findings, it is evident that the MCDA techniques have been applied extensively to assist decision-makers in the flood and drought management processes. It is worth highlighting that there is an enormous growth in the number of publications from 2015 onward. This might be linked to the United Nations Sustainable Development Goals (SDG 2030), set in 2015; however, there is no study establishing or verifying this link.

The distribution of the articles is shown in Table 5, where rows represent different sets of keywords used for search while columns represent the years of publication. The table highlights that KW7 and KW8 did not retrieve any relevant articles for this study. Please note that these keywords did return a number of articles, but all were filtered out after running through the selection and eligibility criteria. This is perhaps due to the use of the generic term "natural disaster" in KW7 and KW8, which returned a high number of articles but was too broad in the sense that it might discuss some other natural disaster instead of flood or drought. In some cases, the article was filtered out because it was not discussing a specific case study. Nevertheless, it is still subjective and open to further investigation of the factors that might influence MCDA applications for flood and drought management.

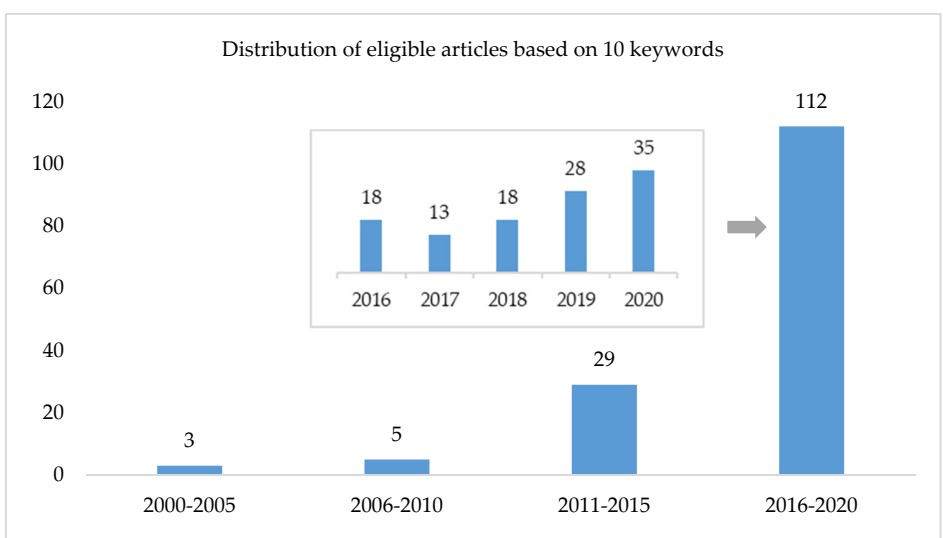

**Figure 3.** Distribution of the shortlisted 149 articles based on ten keywords (Eligibility Phase).

**Table 5.** Distribution of eligible articles based on search keywords.

| | 2004 | 2005 | 2006 | 2007 | 2008 | 2009 | 2010 | 2011 | 2012 | 2013 | 2014 | 2015 | 2016 | 2017 | 2018 | 2019 | 2020 |
|---|---|---|---|---|---|---|---|---|---|---|---|---|---|---|---|---|---|
| KW1 | [32] | [33] | | | | | | | | | | | | [34,35] | [36,37] | [38–40] | [41–52] |
| KW2 | | | | | | | | [53] | | | | [54] | [55] | | | [56] | [57–65] |
| KW3 | | | | | | | | | | | | [66] | [67] | | | [68] | [69] |
| KW4 | | [70] | | | | | | | | [71] | | [72] | | | | | |
| KW5 | | | | | | | | | | [73] | | [74] | [75] | [76,77] | [78] | [79] | [80] |
| KW6 | | | | | | | | | | | | [81] | [82] | [83] | | [84] | [85] |
| KW7 | | | | | | | | | | | | | | | | | |
| KW8 | | | | | | | | | | | | | | | | | |
| KW9 | | | [86] | [87] | | [88] | | | | [89,90] | [91] | [92,93] | [94,95] | [96] | [97–100] | [101–105] | [106–112] |
| KW10 | | | | | [113] | [114] | | [115] | | [116] | [117,118] | [119–130] | [131–142] | [143–149] | [150–160] | [161–176] | [177–180] |

### 4.2. Trends of Articles Based on WRD and DMP

This study shows that 87% (129 articles), as shown in Table 6, focused on applying MCDA techniques in flood management events compared to droughts. The number of flood events recorded, together with the potential increase and reoccurrence of flood events in the future, might contribute to the selection of disaster events in previous studies. There are a great number of studies on the mitigation phase, with 70% (104) of the articles focusing on this phase (as shown in Table 7). There is still a lack of literature on MCDA techniques for both disaster events in other DMP phases (preparedness, response, and recovery). In this paper, the findings might be influenced by the results and outcome of mitigation action, which affected the application of the MCDA technique in other DMP phases in reducing and minimizing the disaster risks and impacts.

**Table 6.** Categorization of articles based on WRD events.

| WRD | No. of Articles | Articles |
|---|---|---|
| Flood | 129 | [32–65,86–180] |
| Drought | 17 | [66–70,73,74,76–85] |
| Drought and Flood | 3 | [71,72,75] |

**Table 7.** Categorization of articles based on DMP.

| DMP Phase | No. of Articles | Articles |
|---|---|---|
| Mitigation | 104 | [33–35,40,41,45,49–52,54–58,60–65,70,71,73,74,76–78,80,82–91,94,95,97–100,102,103,108,109,111–114,116–128,130,133,136–138,141,143–152,154–157,159–163,165–169,171,172,174,175,177–179] |
| Preparedness | 30 | [36,38,39,43,44,46–48,53,61,66,67,69,79,81,104–107,110,129,131,132,134,135,139,140,142,164,170] |
| Response | 13 | [32,37,72,75,92,93,96,101,115,158,173,176,180] |
| Recovery | 2 | [68,153] |

Although there is no correct sequence of phases to be followed in the DMP, understanding the requirements and impacts of each phase is important. It is imperative to consider the importance of each phase equally to support decision-makers in the management of flood and drought events. Even though mitigation is the most prominent phase focused on by decision-makers, it can be argued that studying other DMP phases (preparedness, response, and recovery) is also important and might improve the decision-making processes for managing these disasters. Therefore, MCDA applications in other DMP phases are discussed separately in Section 5.1 in detail.

*4.3. MCDA Technique for WRD Events*

Based on the analysis shown in Figure 4, the application of a single MCDA technique is more preferred compared to the combination of multiple techniques. This paper has identified AHP, mixed-methods, and TOPSIS as the top three MCDA techniques applied to manage floods and droughts. These techniques represent 80% (120) of articles, which indicates that the strengths and weaknesses of the MCDA techniques might influence the selection of MCDA techniques in flood and drought management. For instance, numerous studies use the AHP technique, and it can be argued that this generates a herd behavior where other researchers also tend to choose the AHP technique due to its wide use. This will allow replication and expansion of this technique in other studies.

Meanwhile, mixed-method techniques are attracting attention due to their ability to improve and better justify the decisions related to managing flood and drought events. However, identifying a correct combination among MCDA techniques for the mixed-method approach would be an interesting area to investigate further.

Table 8 shows the classification of MCDA techniques based on WRD events. Details on the distribution pattern for AHP, mixed-methods, and TOPSIS techniques based on WRD for the last 20 years are shown in Figure 5. Due to the higher frequency of events and their severity, floods are the most researched disaster event, resulting in a lack of references on the MCDA technique for drought management. It is interesting to explore two potential areas of study from this finding. Firstly, the study of the viability of other MCDA techniques such as ELECTRE, NAIADE, and CBD in flood and drought management (feasibility and compatibility study) and the possibility of utilizing the same flood management criteria in drought management will benefit countries experiencing both extreme events in terms of data (availability, quantity, and quality), time (data collection, process, and analyzing), and cost associated with managing flood and drought events. Secondly, if further research on the application of the MCDA techniques for drought is undertaken, it could be beneficial in managing the disasters.

*4.4. MCDA Application Based on DMP*

Table 9 shows the MCDA techniques' mapping according to the DMP phases and events in this study. The findings show that the top three MCDA techniques identified in this study focus on the mitigation phase compared to other phases. Out of 104 articles on mitigation, 43 articles were related to AHP and 8 were related to TOPSIS. Interestingly, 21 of these articles discussed the use of mixed-methods. There is a significant gap in MCDA applications between mitigation among other DMP phases (preparedness, response, and

recovery). It can be concluded that mitigation has been receiving attention from decision-makers as part of long-term strategic planning in managing these disasters.

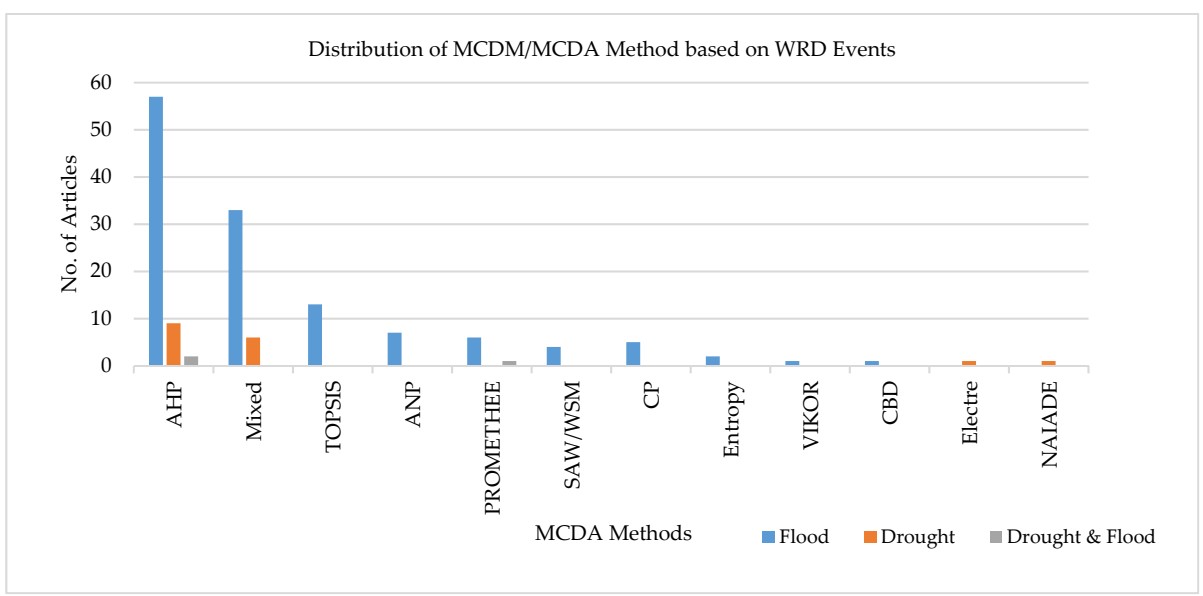

**Figure 4.** Distribution of MCDA techniques based on WRD events.

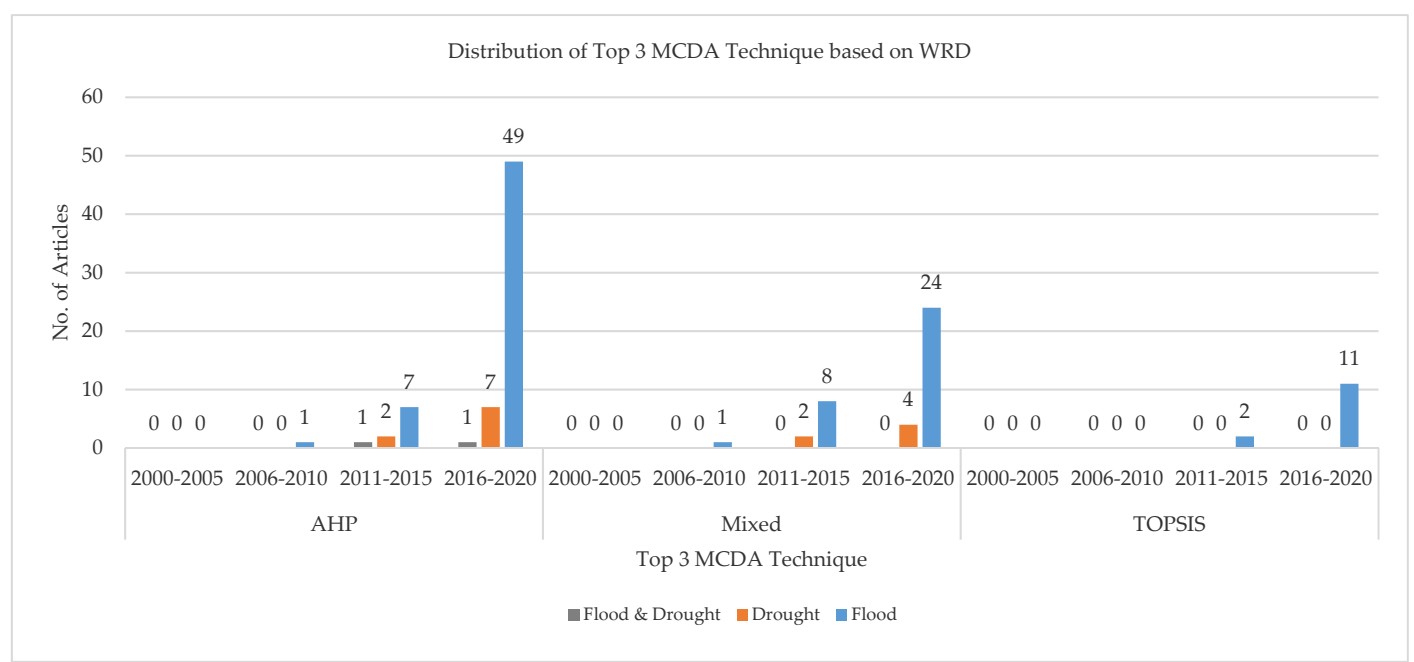

**Figure 5.** Distribution of top three MCDA techniques based on WRD.

In the DMP context, it is questionable whether the MCDA techniques should be implemented in all DMP phases as a structured and cohesive approach to improve overall decisions in disaster management. Although there is a lack of studies conducted on other phases, it is important to explore the possibility of integrating MCDA in other phases of DMP to make a collective and comprehensive disaster management plan. Equal attention should be given to all DMP phases for better planning and strategizing. Decision-makers should explore the potential of incorporating the MCDA technique into each phase of DMP in order to improve the decision results.

**Table 8.** Classification of MCDA techniques based on WRD events.

| MCDA Technique | No. of Relevant Articles | Flood | Drought | Drought and Flood |
|---|---|---|---|---|
| Analytic hierarchy process (AHP) | Flood: 57 Drought: 9 Drought and Flood: 2 | [40,47,48,51,52,55,57–59,61,62, 90,95,101,106,108,109,111– 113,115,117,121,124,125,127, 131,133,135,136,139,141,144– 149,151–154,157–159,161– 164,166,167,170–172,178–180] | [66,76–81,84,85] | [71,75] |
| Mixed-methods | Flood: 33 Drought: 6 Drought and Flood: 0 | [34,36,38,42–46,49,50,53,63– 65,88,89,93,97,98,102,105,107, 122,123,128– 130,140,156,160,165,176,177] | [67–69,73,74,83] | - |
| Technique for the order of prioritization by similarity to ideal solution (TOPSIS) | Flood: 13 Drought: 0 Drought and Flood: 0 | [39,41,91,94,96,99,100,116,134, 138,142,155,173] | - | - |
| Analytic network process (ANP) | Flood: 7 Drought: 0 Drought and Flood: 0 | [33,37,87,103,104,110,174] | - | - |
| Preference ranking organization method for enrichment of evaluations (PROMETHEE) | Flood: 6 Drought: 0 Drought and Flood: 1 | [35,60,118,126,143,175] | - | [72] |
| Compromise programming (CP) | Flood: 5 Drought: 0 Drought and Flood: 0 | [56,86,114,119,120,132] | - | - |
| Simple additive weighting (SAW), or weighted sum model (WSM) | Flood: 4 Drought: 0 Drought and Flood: 0 | [32,54,137,150] | - | - |
| Entropy | Flood: 2 Drought: 0 Drought and Flood: 0 | [168,169] | - | - |
| Choose by disadvantages (CBD) | Flood: 1 Drought: 0 Drought and Flood: 0 | [56,86,114,119,120,132] | - | - |
| VIKOR | Flood: 1 Drought: 0 Drought and Flood: 0 | [92] | - | - |
| Elimination and choice translating reality (ELECTRE) | Flood: 0 Drought: 1 Drought and Flood: 0 | - | [82] | - |
| Novel approach to imprecise assessment and decision environment (NAIADE) | Flood: 0 Drought: 1 Drought and Flood: 0 | - | [67–70,73,74,83] | - |

From criteria identification and selection perspectives, the concept of data analytics, especially on big data, could be further explored to understand the impact (quantity vs. quality) of numerous criteria on the MCDA results. New data creation such as projection data, social media data, structured and unstructured data might influence decision-makers to identify and select criteria in MCDA technique application for managing disaster events.

*4.5. Application of MCDA Mixed-Method Techniques*

Although the most common and popular MCDA techniques are single techniques such as AHP and TOPSIS, mixed-method techniques have been becoming prominent in

disaster management applications in the last five years, as shown in Table 10. In this study, 39 out of 149 articles discussed the MCDA mixed-method technique application, where 61% (24 articles) of these mixed-method technique articles focus on the mitigation phase. Table 11 shows the mapping of mixed-method techniques in the mitigation phase, while Table 12 shows the mapping of mixed-method techniques applications for preparedness, recovery, and response phases.

**Table 9.** Overall classification of MCDA techniques based on WRD events and DMP.

| MCDA Technique | Flood | | | | Drought | | | | Drought and Flood | | | |
|---|---|---|---|---|---|---|---|---|---|---|---|---|
| | MIT. | PREP. | REC. | RESP. | MIT. | PREP. | REC. | RESP. | MIT. | PREP. | REC. | RESP. |
| AHP | [40,51,52,55,57, 58,61,62,90,95, 108,109,111– 113,117,121,124, 125,127,133,136, 141,144– 149,151,152,154, 157,159,161– 163,166,167,171, 172,178,179] | [47,48,59,106, 131,135,139, 164,170] | [153] | [101,115, 158,180] | [76–78,80,84,85] | [66,79,81] | - | - | [71] | - | - | [75] |
| Mixed methods | [34,42,45,49,50, 63–65,88,89,97, 98,102,122,123, 128,130,156,160, 165,177] | [36,38,43,44, 46,53,105,107, 129,140] | - | [93,176] | [73,74,83] | [67,69] | [68] | - | - | - | - | - |
| TOPSIS | [41,91,94,99,100, 116,138,155] | [39,134,142] | - | [96,173] | - | - | - | - | - | - | - | - |
| ANP | [33,87,103,174] | [104,110] | - | [37] | - | - | - | - | - | - | - | - |
| CBD | [56] | - | - | - | - | - | - | - | - | - | - | - |
| CP | [86,114,119,120] | [132] | - | - | - | - | - | - | - | - | - | - |
| ELECTRE | - | - | - | - | [82] | - | - | - | - | - | - | - |
| Entropy | [168,169] | - | - | - | - | - | - | - | - | - | - | - |
| NAIADE | - | - | - | - | [70] | - | - | - | - | - | - | - |
| PROMETHEE | [35,60,118,126, 143,175] | - | - | - | - | - | - | - | - | - | - | [72] |
| SAW/WSM | [54,137,150] | - | - | [32] | - | - | - | - | - | - | - | - |
| VIKOR | - | - | - | [92] | - | - | - | - | - | - | - | - |

Note: MIT—mitigation; PREP—preparedness; REC—recovery; RESP—response.

**Table 10.** Distribution of MCDA mixed-method techniques from 2000 to 2020.

| Year | 2000–2005 | 2006–2009 | 2010–2015 | 2016–2020 |
|---|---|---|---|---|
| No. of Articles | 0 | 1 | 10 | 28 |

**Table 11.** Application of mixed-method techniques in the mitigation phase.

| No. | DMP Phase | WRD | Article | AHP | TOPSIS | SAW/WSM | VIKOR | ANP | ENTROPY | ELECTRE | CP | SWARA | WLC | PROMETHEE | MAUT | SMAA | WPM | REGIME | EVAMIX | Catastrophe | BWM |
|---|---|---|---|---|---|---|---|---|---|---|---|---|---|---|---|---|---|---|---|---|---|
| 1 | Mitigation | Flood | [160] | x | x | | | | | | | | | | | x | | | | | |
| 2 | | | [88] | x | | | | | | x | x | | | | | | | x | x | | |
| 3 | | | [89] | | x | x | | | | | | | | | | | | | | | |
| 4 | | | [97] | x | x | | | x | | | | | | | | | | | | | |
| 5 | | | [98] | x | | | | x | | | | | | | | | | | | | |
| 6 | | | [102] | x | x | | x | | | | | | | | | | | | | | |
| 7 | | | [165] | | x | x | x | | | | | | | | | | | | | | |
| 8 | | | [122] | x | x | x | x | | | x | x | | | | | | | | | | |
| 9 | | | [123] | x | | | | | | | | | | | x | | | | | | |
| 10 | | | [177] | | | | x | | x | x | | x | | x | | | | | | | |
| 11 | | | [128] | x | x | | | | | | | | | | | | | | | | |
| 12 | | | [156] | | | x | | | | | | | | | | | x | | | | |
| 13 | | | [34] | | x | | | | x | | | | | | | | | | | | |
| 14 | | | [130] | | x | x | | | | | | | | | | | | | | | |
| 15 | | | [42] | x | | | | x | | | | | | | | | | | | | |
| 16 | | | [45] | x | | | | | | | | | x | | | | | | | | |
| 17 | | | [49] | | | | | | | | | x | | | | | | | | | x |
| 18 | | | [63] | x | | | | x | | | | | | | | | | | | | |
| 19 | | | [64] | x | | | | | | | | | x | | | | | | | | |
| 20 | | | [50] | | x | x | | | | | x | | | | | | | | | | |
| 21 | | | [65] | | | | | | x | | | | | | | | | | | x | |
| 22 | | Drought | [73] | x | x | | | | | | | | | | | | | | | | |
| 23 | | | [74] | | x | x | | | | | | | | | | | | | | | |
| 24 | | | [83] | x | | | | x | | | | | | | | | | | | | |

**Table 12.** Application of mixed-method techniques for preparedness, recovery, and response phases.

| No. | DMP Phase | WRD | Article | AHP | TOPSIS | SAW/WSM | PROMETHEE | CP | ELECTRE | VIKOR | ANP | DEMANTEL | OWA | WPM | MAUT | CRITIC |
|---|---|---|---|---|---|---|---|---|---|---|---|---|---|---|---|---|
| 1 | Preparedness | Flood | [38] | x | | | x | | | | | | | | | |
| 2 | | | [140] | x | | | | x | | | | | | | | |
| 3 | | | [53] | x | x | x | | x | | | | | x | x | | |
| 4 | | | [105] | x | x | | | | | | | | | | | |
| 5 | | | [129] | | | x | x | | | | | | | | | |
| 6 | | | [36] | x | | | x | | | | | | | | | |
| 7 | | | [107] | | | | | | | | | x | x | | | |
| 8 | | | [43] | x | | x | | | | | | | | | | |
| 9 | | | [44] | | x | x | | | x | x | | | | | | |
| 10 | | | [46] | x | | | x | | | | | | | | | |
| 11 | | Drought | [67] | x | | | x | | | | | | | | | |
| 12 | | | [69] | | x | x | | | | | | | | | | |
| 8 | Recovery | Drought | [68] | | | | | | | | | | | | x | x |
| 9 | Response | Flood | [176] | | x | x | | | | | | | | | | |
| 10 | | | [93] | x | x | x | | x | x | x | | | | | | |

The findings suggest that the mixed-method techniques help address the limitations of a single method. For example, the techniques of AHP [93], TOPSIS [156], and VIKOR [165] had been used individually in the past for investigating flood hazard susceptibility in order to identify flood-prone areas, but recently, Arabameri et al. [103] proposed the use of mixed-method techniques for this purpose. They used AHP to weight the importance of criteria and applied TOPSIS and VIKOR to assess the flood-prone areas [103]. Another example is the use of SAW/WSM for criteria weighting while using PROMETHEE to evaluate and rank the criteria in determining the optimal measures for flood risk [130].

In a mixed-method technique, a combination of the MCDA techniques could be explored further. Although AHP is the most commonly used technique to combine with other

techniques, other techniques such as PROMETHEE, MAUT, and composite programming were also used in the mitigation phase. We also found that ANP, CRITIC, and WPM were rarely applied in preparedness, response, and recovery. Therefore, in applying the mixed-method technique, decision-makers need to consider suitable combinations of techniques to ensure the best results based on mixed methods. It is also important to understand the practicality of using a mixed-method technique based on criteria perspectives such as relevance and uncertainty, data availability, and stakeholder and expert opinions.

There were few mixed-method technique applications in other DMP phases. Decision-makers should consider introducing and applying the MCDA techniques in other DMP phases. This approach will provide a systematic solution in each step of the DMP to improve the overall decision-making outcome.

*4.6. Criteria Selection in MCDA Application*

As MCDA considers multiple criteria, criteria identification and selection are important to support decision-making. In this study, the number of criteria used was found to range from 2 to 54, where for flood events, the minimum number of criteria used was 2, and the maximum was 54. Meanwhile, for drought events, the minimum number of criteria was 3, and the maximum was 17. Based on the number of criteria used in MCDA, criteria quantity and quality (relevance, certainty, and uncertainty) could affect the decision result; thus, decision-makers must identify and select the most relevant criteria to be applied in MCDA.

Types of criteria are also critical based on decision problems. In this study, generally, most of the criteria were selected from hydrology or environment domains such as rainfall, land use, river network, and water storage. Concurrently, other criteria such as population density, gender, age, education, disaster policy, disaster loss, gross domestic product (GDP), economic activities, and technology have also been used for flood and drought management.

Due to various criteria being covered in managing these disasters, criteria identification and selection based on macrodomain and microdomain analysis for each disaster would improve the selection of criteria in terms of quantity and quality of criteria. Thus, it would ensure that the decision result is more efficient and effective in managing the disasters.

## 5. Discussion and Research Opportunities

The findings discussed in Section 4 show that most MCDA techniques have been applied to manage flood events. The high number of flood events recorded and their impacts may influence the pattern of MCDA techniques in this study. The selection of the MCDA technique may affect the decision results. Therefore, understanding the strengths and weaknesses of these MCDA techniques should help decision-makers improve the quality of their decisions. This study has provided an overview of the use of the MCDA techniques for managing flood and drought events, intending to enhance the understanding and exposure of MCDA techniques for concerned decision-makers.

Based on DMP phases, the findings highlighted MCDA techniques' application concentrating on the mitigation phase compared to other phases. As mitigation focuses on prevention for long-term planning, this phase is significant to assist decision-makers in developing comprehensive and collective strategic planning. In contrast, preparedness, response, and recovery are focused on short-term planning, which focuses more on operational planning. It can be argued that the lengthy process of structuring decision problems is only suitable for long-term planning. In contrast, most short-term planning decisions might have deadlines and other constraints that end up in making quick and intuitive decisions. On the contrary, it can also be argued that these short-term planning decisions can be improved by developing a process based on historical data on previous similar decisions and creating MCDA structures based on these historical data. This will make MCDA techniques useful even for situations where time is of critical importance. Considering

the latter argument, disaster management can be improved with MCDA techniques for strategic and operational planning.

*5.1. Research Opportunities*

The findings show a lack of study on several areas of MCDA application for flood and drought management. First of all, only a handful of MCDA techniques have been applied in this area, leaving a number of other techniques still to be explored and used. Then there is an issue of using only a handful of criteria in these case studies; that is, most of the studies were limited to the environmental and social factors (or criteria). It can be argued that these case studies should be revisited by considering other important factors such as economic, technical, and political criteria. Another important finding is that most of the studies focused on the mitigation phase of DMP. The other three phases were relatively less explored and studied within the context of MCDA applications. Last but not least, our analysis suggests that case studies on drought events are almost nonexistent, which is an important gap to be filled in this area.

To summarize, four areas of study can be considered as opportunities for future research, namely (a) using other MCDA techniques, (b) considering more criteria, (c) using MCDA in other DMP phases, and (d) investigating drought events. These four areas are discussed below in more detail.

5.1.1. Using Novel MCDA Techniques

Although AHP, mixed-method techniques, and TOPSIS have dominated the types of MCDA techniques used for flood and drought disasters, further studies should be conducted to explore the possibility of other techniques that might improve decision results. In the mixed-method techniques, the feasibility and compatibility of other techniques to be combined together would be an interesting area to be explored. This will allow the decision-makers to see the effectiveness of combined techniques in the quality of the decision results. Regardless of either single or mixed-methods techniques being applied, a comparative study can continuously be conducted to see the resulting effectiveness based on different MCDA techniques. For instance, understanding the selection of MCDA technique (what and why) in flood or drought management using the MCDA approach could be further explored to understand how MCDA could improve decision results in flood and drought management.

Table 13 summarizes the pros and cons of using different MCDA techniques in managing water-related disasters. The table rates each method against a list of important characteristics. There are ✓ and × symbols used to highlight whether the method supports or doesn't support these characteristics. For example, looking at the first row and first column, AHP is considered a suitable method when communicating to non-technical people, however, looking at the next column, ANP is considered an unsuitable method for this purpose. Please note that the boldified tick symbol ✓ is used to emphasize on methods that are highly suitable. For example, in the second row, AHP, ANP and SURE are highly suitable methods for situations where inconsistencies should be allowed in human judgments. The table might help relevant decision experts to choose the most appropriate MCDA techniques for their decision problem in their environment and context.

5.1.2. Considering More Criteria

Selecting criteria to be applied in MCDA required a thorough investigation. Selecting relevant and rightful criteria would affect the decision result; thus, it is important to identify and select the suitable criteria for flood and drought disaster management. For instance, a study to understand the relevance criteria for flood and drought management could expand decision-makers' options for criteria selection in the MCDA approach. The number of criteria should be considered, and data that would support the criteria also need to be inspected. The criteria quality and quantity used in MCDA can be studied further from both a microdomain and macrodomain viewpoint. Criteria analysis using analysis models

such as strengths, weaknesses, opportunities, and threats (SWOT) or political, economic, social, technological, environmental, and legal (PESTEL) could be used to identify the most relevant and suitable criteria based on expert's opinion and judgment and finite data availability.

**Table 13.** Pros and cons of using different MCDA techniques in managing water-related disasters.

| | AHP | ANP | DEA | WSM | WPM | GP | ELECTRE | Grey | MAUT | CBR | SMART | PROMETHEE | TOPSIS | SURE |
|---|---|---|---|---|---|---|---|---|---|---|---|---|---|---|
| Communicating to nontechnical people | ✓ | × | × | ✓ | . | × | × | . | ✓ | . | ✓ | × | ✓ | ✓ |
| Allows inconsistencies in human judgments | ✓ | ✓ | . | . | . | . | . | ✓ | . | ✓ | . | . | . | ✓ |
| Robust against rank reversal | × | . | . | × | . | . | ✓ | . | . | . | × | . | . | × |
| Criteria can have different units of measurement | ✓ | ✓ | ✓ | × | ✓ | ✓ | ✓ | ✓ | ✓ | ✓ | . | ✓ | . | ✓ |
| Takes uncertainty into account | . | . | . | . | . | . | . | ✓ | . | ✓ | . | . | . | ✓ |
| Supports indifference and vetoes | × | × | . | × | × | . | ✓ | . | × | × | × | ✓ | . | × |
| One criterion compensates for others | ✓ | ✓ | . | ✓ | ✓ | . | × | . | ✓ | . | ✓ | × | ✓ | ✓ |
| Robust against the trap of averages | × | × | . | × | × | ✓ | ✓ | . | × | ✓ | × | ✓ | ✓ | × |
| Easier to compute | × | × | . | ✓ | ✓ | . | × | . | ✓ | . | ✓ | × | ✓ | . |
| Can be applied to any size of problem | × | × | ✓ | ✓ | ✓ | ✓ | × | ✓ | ✓ | × | ✓ | . | ✓ | ✓ |
| Can adapt to slight changes | ✓ | ✓ | . | . | . | . | . | ✓ | . | ✓ | . | . | . | . |
| Can be supported with visual aid | ✓ | . | ✓ | ✓ | ✓ | . | × | . | . | × | ✓ | ✓ | ✓ | ✓ |

### 5.1.3. Considering Other DMP Phases

MCDA applications in the DMP phases focused more on the mitigation phase for both disaster events. Future research could be conducted on MCDA application in other DMP phases as a structured disaster management approach. A study on uniformity for criteria analysis technique might be applied in every phase of DMP to improve the decision result regardless of the MCDA technique. With a homogeneous criteria analysis technique applied throughout the DMP phases, it could be an interesting area to investigate how MCDA could impact the decision result in disaster management. For example, a comparative and feasibility study could be conducted to analyze and identify effective and suitable techniques for managing floods or droughts in all phases of DMP. This would give the decision-makers a basis to decide the potential MCDA technique to be applied in flood and drought management based on the availability of finite datasets, opinions, and stakeholders' judgments.

### 5.1.4. Investigating Drought Events

While flood events are gaining more attention as a case study in disaster management, the focus on MCDA for drought events also can be expanded for future reference by other studies. Replication and extension of MCDA for flood events could be applied to drought events, such as selecting and ranking drought-prone areas, developing a drought risk map, and setting criteria of drought vulnerability. Since drought and flood will mostly be using the same criteria, the study on drought events could be conducted with slight adjustments to criteria such as pattern changes in rainfall and temperature and alternative water (groundwater alternative). Despite a lack of studies focusing on drought events, there are opportunities to explore drought management based on the other phases of DMP. For example, a potential study of MCDA techniques such as PROMETHEE, ELECTRE, compromise programming, or ANP in managing drought events for preparedness, response, and recovery phase could be investigated further on the feasibility and effectiveness of MCDA techniques. These studies would be of interest to academics working in decision-making, policymaking and governance, MCDA techniques, environment, or engineering areas.

Adaptation of MCDA techniques may support decision-makers in both operational and strategic planning. MCDA techniques may be applied directly or indirectly in the

context of DMP for structural or nonstructural measures. Table 14 summarizes the possible measures (structural and nonstructural) identified in this study. The examples of the MCDA type of problem shown in Table 14 give an idea and overview of how the MCDA techniques can be implemented to solve weightage, ranking, and prioritizing the type of problem for managing flood and drought events. The possible measures listed in Table 14 are not finite; further study on other possible measures is required, which involving policymakers, hydrology, environment, social, and economic experts to produce and develop more constructive flood and drought measures. This will help decision-makers to improve the decision to reduce risks and impacts of these disasters in the future.

**Table 14.** Possible measures according to DMP.

| No. | DMP Phase | Probable Flood and Drought Measures | Example of Type MCDA Problem |
|---|---|---|---|
| 1 | Mitigation | Flood and drought hazard mapping | 1. To choose flood and drought plain maps to be used for recovery action;<br>2. To sort the best-suitable flood and drought maps to be used for long term planning;<br>3. To rank the flood and drought maps based on criteria set by decision-makers. |
| | | Possible advanced and strategic planning (example: forecasting, projection, prediction, and real-time disaster information collection) | 1. To choose the best-fit forecasting data to be used in the development of inundation and hazard maps;<br>2. To select the most suitable developed hazard and inundation maps to be used in managing disaster events;<br>3. To rank the best possible projection data to be used in index assessment;<br>4. To rank the best-fit hazard map to be implemented. |
| | | Possible disaster risk and reduction assessment Examples:<br>1. Flood/drought vulnerability index;<br>2. Flood/drought inundation maps;<br>3. Flood/drought readiness index;<br>4. Flood/drought adaptation index and hazards maps). | To choose, sort, rank, and describe criteria, factors, indicators, and parameters to be used in conducting the assessment |
| | | Development of zoning map | To choose, sort, and rank the zoning maps for operational and strategic planning (short term and long-term plan) |
| 2 | Preparedness | Future-looking scenarios to plan | Choice of data to be used in developing the future-looking scenarios |
| | | Natural disaster insurance—the incentives provided should be appealing and disseminated for homeowners and businesses | 1. Choice and rank of the area of high risk to be covered;<br>2. Choice and rank of insured to be covered for preparedness action. |
| | | Building awareness, education, and capacity-building culture around risk | Choice and selection of the high-risk area to conduct the program |
| | | Setting up an evacuation place | Choice and selection of highly recommended areas to build evacuation place based on criteria sets |
| 3 | Response | Warning, evacuation, and search and rescue | To select and rank high-risk location earlier response plan |
| | | Immediate assistance, loss, and damage assessment, | To choose, select, and rank most vulnerable areas or communities that need an immediate response from authority bodies |
| 4 | Recovery | Plan and policy adaptation (financial and nonfinancial) to increase the resilience to WRD | 1. To choose and sort significant factors to be incorporated into policy development;<br>2. To rank recovery action required based on the WRD disaster impacts (prioritizing recovery actions to be taken). |
| | | Assessment for the reconstruction of repeat loss infrastructure and properties—buy-out or mitigate | 1. To choose and sort identified infrastructure for reconstruction based on a set of factors;<br>2. To rank the most impacted infrastructure worth being reconstructed. |
| | | Up-front option and plan information | To choose, sort, and rank significant and relevant options and plans to be implemented based on WRD disasters' impact |

## 6. Conclusions

By investigating MCDA techniques for managing flood and drought events in the 21st century, we have identified an increasing trend in applications. By referring to articles published in journals, conferences, and proceeding papers, this paper reviewed 149 articles on MCDA techniques for flood and drought management published between 2000 and 2020. The substantial impacts of the flood events have attracted more studies and have become the most focused events studied by decision-makers in the DMP. Mitigation is the widest phase used for strategic and operational planning, with AHP being the most commonly used MCDA technique.

There is a significant gap in the number of other MCDA techniques applied to manage drought and flood events. The application of these techniques also shows a substantial gap according to DMP phases. With the current impacts of climate change, which increases the frequency and magnitude of flood and drought events, studies on MCDA applications could be explored further.

According to the findings and discussion (Sections 4 and 5), the MCDA technique, criteria selection, application according to DMP phase, and the type of disaster event (in this study, drought) are all potential areas for further study to extend the results and findings available in the literature. The findings of this paper suggest the importance of using other MCDA techniques to enhance flood and drought disaster management, either as a single approach or as a combination of MCDA techniques in a mixed-method approach. The analyses on the MCDA technique selection could be beneficial in MCDA application to improve the decision results based on the technique's effectiveness in solving the problem. Criteria identification and selection is an important component to support decision-making; thus, strategic analysis can be undertaken to assist decision-makers in focusing on pivotal criteria that are relevant and significant to improve the management of flood and drought events.

An exploratory study on applying the MCDA technique as a uniformity approach in each phase of the DMP would offer a more structured mechanism in the decision-making process, potentially having substantial effects on decision results. More studies on applying MCDA techniques in drought disasters are needed. An analysis of the MCDA technique for drought events would provide additional information to assist decision-makers in selecting and strategizing on the MCDA technique and applying the MCDA technique in the DMP phases and criteria selection.

While addressing the various topics of potential study interest, this paper provides valuable and significant information to justify and recommend prospective studies. Exploring these areas will enhance macro- and micro-environmental decision-making by considering all criteria and factors in flood and drought management.

**Author Contributions:** Conceptualization, M.F.A.; methodology, M.F.A.; validation, S.S. and R.E.H. formal analysis, M.F.A.; investigation, M.F.A., S.S. and R.E.H.; resources, M.F.A.; data curation, M.F.A.; writing—original draft preparation, M.F.A.; writing—review and editing, S.S. and R.E.H.; supervision, S.S. and R.E.H. All authors have read and agreed to the published version of the manuscript.

**Funding:** This research was funded by Public Service Department of Malaysia.

**Institutional Review Board Statement:** Not applicable.

**Informed Consent Statement:** Not applicable.

**Data Availability Statement:** The data presented in this study are available on request from the corresponding author. The data are not publicly available due to on-going study.

**Acknowledgments:** This work was supported by National Water Research Institute of Malaysia (NAHRIM). Thank you to the researchers and management of NAHRIM for their useful input.

**Conflicts of Interest:** The authors declare no conflict of interest.

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
