# Peer review of "An Overview of Multi-Criteria Decision Analysis (MCDA) Application in Managing Water-Related Disaster Events: Analyzing 20 Years of Literature for Flood and Drought Events"

_water, doi:10.3390/w13101358_

Round 1

Reviewer 1 Report

The authors re-edited the title of the paper. The present title is more consistent with the content and does not carry a load of unnecessary readers' expectations. In a letter to the reviewer, the authors claim that they have moved from a statistical summary of the literature/researches to their critical analysis. I am reluctant to agree with this statement. I haven’t noticed any departure from the statistics in the text. This part of the paper has remained almost unchanged. However, the authors significantly improved the background section. For example, the addition of Table 3 strengthened the 2.3 section on MCDA much. The same applies to discussion section. In my opinion, in its current form, the paper contains valuable information or  guidance that may be of assistance to decision-makers in solving problems before, during, and after the disaster.

However, the re-edited abstract raises my doubt. I recommend not to use in the abstract the phrases like: "The object of this paper are: to provide...".  It would be enough to say: "The paper provides...”. Lines 19-21, in my opinion, are too detailed and unnecessary.

Reviewer 2 Report

The authors addressed all the comments from the first review and improved the article content. 

Reviewer 3 Report

Please follow the literature review with a clear and concise state of the art analysis. This should clearly show the knowledge gaps identified and link them to your paper goals. Please reason both the novelty and the relevance of your paper goals. It is difficult to read the Figure 1 (PRISMA flow diagram for the selection of eligible studies), it is too small. What is the added value of the paper? Conclusions are very general. Why the paper should be recommended? The bibliography is relevant but presents some minor lacks when it comes to citations and mentions. To clarify some aspects, I would suggest that the authors write the bibliography evenly: for example, journal papers require either the complete journal name, or the JCR abbreviation), or the ISO abbreviation. The manuscript is prepared in poor form. It should be improved and prepared according to the journal's guidelines (tables, poor quality of figures, etc.).

Round 2

Reviewer 3 Report

Accept in present form.

This manuscript is a resubmission of an earlier submission. The following is a list of the peer review reports and author responses from that submission.

Round 1

Reviewer 1 Report

The authors announce the paper as a comprehensive review of the applications and trends of using Multi-Criteria Decision Analysis (MCDA) techniques for managing flood and drought events. In my opinion this is not a true.  For me, the paper is an analysis (mostly statistical) of the previous researches using MCDA in water related disasters.   

The only thing the reader gains after reading this paper is an extensive set of literature and Introduction plus Background sections, in which the terms disaster, water-related disaster events, Disaster Management Plan or Multi-Criteria Decision Analysis are presented and explained. However, I get the impression that the advantages of the paper end there.

The Findings section contains only the statistics of the analyzed articles depending on the used division by keywords, WRD or DMP. The main conclusions are such as: "Approximately 89% articles… focused on the application of MCDA techniques in the management of flood events.", or: "The keyword search… shows that there was an upward trend of publication from 2017 to 2019."

So, the article is just a simple report on the uses of MCDA in research on water-related disasters. It does not contain valuable information or guidance that may be of assistance to decision-makers in solving problems before, during, and after the disaster.

Therefore, I do not think this article could be of interest to a larger readership. Therefore, I do not recommend it for publication.

Reviewer 2 Report

This paper is a review paper on the Multi Criteria Decision Analysis (MCDA) for managing water-related disasters, namely flood and drought events. I was very curious about MCDA methods and their application in the hydrological literature before reading the manuscript and I was expecting to learn about these methods by reading the manuscript. However, the manuscript does not propose a classical review of the methods, discussing technicalities about the methods, advantages, disadvantages, fields of application and lessons learned from specific problems/case studies. In fact, the manuscript presents a numerical analysis of the type and number of manuscript dealing with flood and drought events and MCDA that only gives numbers about the implementation of the methods. What I mean is that this review article does not discuss the methods but only their popularity in recent literature. I do not find this kind of review very useful to the scientific community and the potential readers; hence, I would not recommend this manuscript for publication in Water. If the Editor has a different opinion, I suggest the Authors to reformulate the title of the paper in order to let the reader understand quickly what the manuscript really proposes, that is not a “comprehensive review” yet a “popularity assessment” in recent literature.

Reviewer 3 Report

In this article, the authors reviewed the Multi-Criteria Decision Analysis use in
managing natural disasters, such as floods and droughts. The authors provided a resourceful review paper including the published journals from 2000-2020. I have provided few comments below which will help to improve the article quality more. 

It would be more resourceful if authors can show in which regions (or countries) these MCDA methods were applied to solve the different DMP phases?

Could you update all the Figures and Tables for 2020? Since it is March 2021, you should get the full data for 2020.

Line 46-48-> Reference?

Line 43- what is CRED? Please define

Line 44-> “Both events are likely to amplify in frequency and intensity in the future”- why?

Line 55-56-> provides references for each of these.

Line 86-> how many events in 2010 and 2020 separately? Please provide the numbers for both flood and droughts separately. It will provide evidence- at what rate these events are increasing over the year.

Line 150-> mentioned the strength and weakness in 1/2 lines. Also, why researchers around the world used the AHP most?

Line 222-224-> Not clear, need to rewrite

Line 228-231-> could you mention in which area mitigation measured was mainly focused? Such as urban areas, agricultural areas, etc. to reduce property or food loss.

Line 207, 242-> here mixed-method techniques are referred to as integrated method? Provide a clear description

Reviewer 4 Report

The form of the paper is review not article. The review concerns applications and trends of using Multi-Criteria Decision Analysis techniques for managing flood and drought events.

The following remarks should be referred to: In what way the publications were chosen to perform the analysis? Are they indexed in Wos or Scopus? In what way this review will help decision-makers to prioritize criteria based on the objective of planning? What do you mean by the statement, that the selected criteria should fit the decision-makers’ and practitioners’ selected standards in the flood adaptation index calculation? Abstract must be rewritten in a more precise style (The highest number of articles, how many?). It should be expanded to include the important results. Different methods should be compared together, the advantage and disadvantage should be illustrated.

Manuscript is prepared in poor form. It should be improved and prepared according to the journals guidelines (tables, poor quality of figures, etc.). The material has a very low degree of originality.